# Social Media Addiction among Vietnam Youths: Patterns and Correlated Factors

**DOI:** 10.3390/ijerph192114416

**Published:** 2022-11-03

**Authors:** Linh Phuong Doan, Linh Khanh Le, Tham Thi Nguyen, Thao Thi Phuong Nguyen, Minh Ngoc Vu Le, Giang Thu Vu, Carl A. Latkin, Cyrus S. H. Ho, Roger C. M. Ho, Melvyn W. B. Zhang

**Affiliations:** 1Faculty of Medicine, Duy Tan University, Da Nang 550000, Vietnam; 2Institute for Global Health Innovations, Duy Tan University, Da Nang 550000, Vietnam; 3Department of Economics, Troy University, Troy, AL 36082, USA; 4Institute of Health Economics and Technology, Hanoi 100000, Vietnam; 5Center of Excellence in Evidence-Based Medicine, Nguyen Tat Thanh University, Ho Chi Minh City 700000, Vietnam; 6Bloomberg School of Public Health, Johns Hopkins University, Baltimore, MD 21205, USA; 7Department of Psychological Medicine, Yong Loo Lin School of Medicine, National University of Singapore, Singapore 119228, Singapore; 8Institute for Health Innovation and Technology (iHealthtech), National University of Singapore, Singapore 119077, Singapore; 9Lee Kong Chian School of Medicine, Nanyang Technological University Singapore, Singapore 639815, Singapore

**Keywords:** social media addiction, internet disorder, behavioral addiction, Internet, epidemiology

## Abstract

Background: Prior research has investigated how the excessive use of social media has an impact on one’s functioning. Youths are among the most vulnerable subjects to the impacts of social media overuse, especially in Vietnam (a developing country). However, very little evidence has been provided on social media addiction. This work aims to determine the prevalence of social media addiction amongst Vietnamese individuals and the factors associated with social media addiction. Method: An online cross-sectional study was conducted from June to July 2020 in Vietnam. Participants completed the structured questionnaire, which comprised four main components: (1) social characteristics; (2) The Bergen Social Media Addiction Scale; (3) stress associated with neglect and negative reactions by online peers and fear of missing out (FOMO); and (4) status of social media platform usage. Results: Some demographic factors, such as gender, the locality of accommodation, and relationship status affected the overall scores. The results also indicated Facebook, Zalo, and Youtube to be the most popular social media platforms among Vietnamese youths. Individuals who used social media for gaming also had higher BSMAS scores. FOMO and stress associated with neglect by online peers had a high correlation with social media addiction. Conclusions: This study is one of the first studies to examine social media addiction and its associated factors in Vietnam. Interventions for social media addiction need to be developed in different fields: clinical research, policy, and education.

## 1. Introduction

In the last century, technological advances have revolutionized the way we approach life, from small, routine lifestyle habits to the operation of governments or the delivery of healthcare [1]. It is undeniable that technology has made our lives easier, as it provides us with ease of information access, a variety of communication means, productivity increases, and so on, while also being cost-efficient and time-saving [2,3]. Despite numerous benefits, the excessive use of technology has been increasingly recognized as a problem or even a disorder. Recently, the International Classification System for Disorders (ICD-11) officially included gaming disorder as a diagnosable condition [4]. Social media is also increasing in popularity. A recent survey by Global WebIndex highlighted that 53.6% of the population worldwide used social media, and the average daily social media usage was 2 h and 25 min [5]. Other studies, such as that by Kemp in 2020, also reported a total of 4 billion individuals actively using social media as of July 2020 [6]. It is predicted that by 2026, 4.75 billion people will have a social network account, meaning nearly 6 out of every 10 people on the planet will be active online [7].

Although social media addiction has not yet been classified as a diagnostic disorder, nor is it a subset of the recently established diagnostic criteria for gaming disorder, numerous studies have investigated the impacts of excessive social media use on health and bodily function. A previous study by Pantic et al. (2014) indicated that the overuse of social networking sites/media could lead to an increase in depressive signs and symptoms and have a negative influence on one’s self-perception [8]. The development of depressive symptoms from social media might be a direct result of increased screen time and reduced intensity of interpersonal communications with peers and family [9]. The closest diagnostic criteria for social media addiction, to our knowledge, involved a framework proposed by Griffiths et al. (2014) [10], which included the salience of behavior, mood modification, escapism from reality, tolerance, withdrawal, interpersonal conflicts, and relapse. Given how easily accessible social media is and the potential influences of social media usage on one’s psychosocial functioning, researchers need to understand the epidemiology of social media addiction. A landmark meta-analytical paper published in 2021 also examined the prevalence of social media addictions in 32 nations. Studies included were from North America, Western Europe, Eastern Europe, Asia, the Middle East, Africa, and Latin America, where results indicated that the overall prevalence was 31% for collectivist countries, and 14% for individualistic countries [8]. Among groups of social media users, teenagers and youths are the most vulnerable age group; although they are the youngest, teenagers are also the ones who spend the most time online and thereby becoming the ones most exposed to the underlying risks of social media [11]. Common impacts of social media overuse on young people included sleep disruption, reduced attention span, negative self-perception, and cyberbullying [12,13]. While there has been substantial evidence on the use of social media from a health viewpoint, a review of the literature showed that less than 10 studies focused on teenagers and included adolescents in their sample [14]. Alarming figures suggested a need for not only examining the magnitude of such addictive behaviors among youths but also developing appropriate treatment programs that could reduce these rates.

The Bergen Facebook Addiction Scale (BFAS) and its related modified version, the Bergen Social Media Addiction Scale (BSMAS), are among the most common questionnaires used for the assessment of social media addiction [15,16]. The original BFAS was developed primarily to investigate social media addiction specific to Facebook. The original BFAS was then modified and named BSMAS to keep up with the increased number of social network sites and included assessment across different platforms. In a systematic review of BFAS and BSMAS, Duradoni et al. (2020) identified a total of eight prior studies that used the BFAS [17], and another two which used the BSMAS. Although both scales are relatively new, the validation of both has been tested extensively. In terms of the psychometric properties of BSMAS, a trial among an Italian cohort of 769 participants indicated that the tool has had an excellent fit of the model to the data [18]. Subsequent validation studies were undertaken in other cohorts, most recently in Hong Kong and Taiwan, which also yielded favorable results [19]. Since these initial studies, others such as Chen et al. (2020) [20] have examined the time invariance of three questionnaire instruments—the Smartphone Application-Based Addiction Scale (SABAS), the Bergen Social Media Addiction Scale (BSMAS), and the nine-item Internet Gaming Disorder Scale-Short Form (IGDS-SF9)—amongst Chinese students. The authors reported that the results obtained from these scales are time invariant across at least three months. In another recent study by Chang et al. (2022) [21], the authors highlighted how the prevalence rates of social media addiction vary and that those researchers who adopt a more polythetic approach tend to find more consistency in the prevalence rates obtained from different samples. In addition, the adoption of the polythetic schematic approach allows for the identification of more members of the high-risk group. In another validation study by Stanculescu et al. (2021) [22], the psychometric properties of the scale were validated amongst a Romanian sample, and aspects of the diagnostic criteria that were most relevant for the diagnosis of social media addiction amongst Romanian individuals were salience, conflict, withdrawal, and mood modification.

As a developing country, Vietnam’s internet penetration rates are expected to rise to 81.5% by 2025 [23]. Several previous studies investigating the prevalence of internet addictive behaviors amongst Vietnamese youths reported that 21.5% of the participants suffered from internet addiction and that those with internet addiction were more likely to experience difficulty in performing self-care or daily routine tasks and to suffer from psychiatric disorders such as anxiety or depression [24]. Despite the prevalence of such problems in Vietnam, social media addiction has not received sufficient research attention. There has been a gap in research on social media addiction and hence a delay in interventions aimed at tackling its impact, especially on youths. Therefore, our work aims to determine the prevalence of social media addiction amongst Vietnamese individuals and the factors associated with this addiction, such as peers’ endorsement or rejection and the fear of missing out, and to propose interventions accordingly.

## 2. Materials and Methods

### 2.1. Study Setting and Participants

An online cross-sectional study was conducted from June to July 2020 in Vietnam. Individuals were eligible to participate in the study if they were (1) between 16 to 30 years old (based on Youth Law No. 53/2005/QH11 [25] of the Vietnam National Assembly); (2) currently residing in Vietnam and (3) capable of providing informed consent. Participants were recruited from all provinces of Vietnam. The snowball sampling technique was used for the recruitment. A core group of participants, which included leaders of the youth union in different public institutions, companies, and organizations, were invited to complete an online survey. These individuals then invited peers and people in their network to complete the online survey.

To calculate the sample size of this study, we used the formula for estimating a population mean, with the expected mean score of BSMAS among adolescents = 15.24; standard deviation = 4.83 (according to a previous study in Iran, 2017 [1]); confidence level = 95%; and relative precision = 0.05. After calculating, the necessary sample size was 155 participants. To prevent incomplete responses or dropout, 15% of the sample size was added, thus resulting in a total of 179 participants who were invited to participate in the study. At the end of the data collection period, the total number of participants who agreed to participate in this study and completed the questionnaire was 173, with a response rate of 96.6%.

### 2.2. Measurement and Instrument

In this study, SurveyMonkey’s platform (surveymonkey.com) was used to host the online survey questionnaire. This platform is low-cost, user-friendly, easy to implement, and accessible for samples nationwide. The structured questionnaire comprised four other questionnaires: (1) social characteristics; (2) The Bergen Social Media Addiction Scale (BSMAS); (3) stress associated with neglect and negative reactions by online peers and fear of missing out (FOMO scale); and (4) status of social media platform usage. The survey was first piloted on several youths to ensure the cross-cultural validity of the instruments as translated into Vietnamese as well as to test the logical order and text of each question. Then, the revised questionnaire was uploaded to the online survey portal. The data collection began after the online survey system was tested for accuracy, and all technical issues were addressed.

### 2.3. Variables

#### 2.3.1. Outcome Variable

The Bergen Social Media Addiction Scale (BSMAS)

In this study, the BSMAS scale was used to measure the severity of social media addiction among Vietnamese youths. The BSMAS scale consisted of 6 items, each of which was ranked on a Likert-rating scale with options ranging from 1 (very rarely) to 5 (very often) [16]. The total score ranged between 6 and 30, with higher total scores indicating more severe social media addiction. The psychometric properties of the BSMAS have been validated in Italian [18] and English [26] as well as in Hong Kong [20,27] and Iran [16]. In this study, the Cronbach’s alpha for the BSMAS was 0.81.

#### 2.3.2. Covariates

##### Socioeconomic Characteristic

Participants were asked about gender, age, household income (rich/medium/poor), marital status (single/living together as spouse/married/widowed/don’t want to share), education (below high school and high school/college/tertiary and upper); current living location (urban/town/rural or mountain area); current living partner (family/others).

Furthermore, based on age, the participants were grouped into three subgroups: people below 18 years old were grouped into “adolescents”; people 18–24 years old were grouped into “young adults”; and the remaining group was from 25–30 years old [25,28,29].

##### Stress Associated with Neglect and Negative Reactions by Online Peers and Fear of Missing Out

Stress associated with neglect and negative reactions by online peers: This scale included 8 items, divided into 2 domains: stress associated with neglect by other users (SSN) (4 questions) and stress associated with negative reactions by other users (4 questions). For the first part, participants were asked to score the following statements: “I would feel stressed if my posts did not receive comments”, “I would feel stressed if my pictures and videos did not receive comments”, “I would feel stressed if my posts did not receive Likes”, and “I would feel stressed if my pictures and videos did not receive Likes”. For the next part, participants were asked to score the following statements: “I would feel stressed if my posts received negative comments”, “I feel would feel stressed if my pictures or videos received negative comments”, “I would feel stressed if I got kicked out of social media groups”, and “I would feel stressed if I lost friends/followers on social media”. For each question, the score ranged from 1 to 5 (1 = disagree completely, 5 = completely agree); higher scores indicated higher stress levels [30]. The Cronbach’s alpha was 0.95.

Fear of Missing out scale (FOMO): The FOMO scale consisted of 10 questions about fears, worries, and anxiety in adolescents. A total score of 10 items was used to measure FOMO (10 to 50), with higher scores indicating higher FOMO levels. Each question was scored from 1 to 5 (1—not at all true of me, 2—slightly true of me; 3—moderately true of me, 4—very true of me, and 5—extremely true of me) [31]. The Cronbach’s alpha was 0.91.

##### Status of Social Media Platform Usage

This questionnaire comprised four questions: “Do you use social media networks?”, “What was the year you started to use social media networks?”, “How much time do you spend using social media every day?” (in hours) and “What is your purpose for using social media?”.

### 2.4. Data Analysis

Both descriptive and analytical statistics were used by STATA version 16. Continuous variables were presented as mean and standard deviation (SD), while categorical variables were presented as frequencies with percentages. To compare differences between the status of social media addiction among youths and some characteristics, we used the Wilcoxon rank-sum test, Kruskal–Wallis tests for continuous variables, and χ^2^. A *p*-value (*p*) < 0.05 was considered statistically significant. Tobit-censored regression was used to determine factors related to the BSMAS scale.

Construct validity: Based on the observed data, exploratory factor analysis (EFA) with principal component analysis (PCA) was utilized to identify the instrument’s ideal structural model. The number of components was calculated using the scree plot and parallel analysis; eigenvalues and the amount of variance were explained. Items with a loading value ≥ 0.4 were considered to be included in the relevant component [32].

Then, Confirmation Factor Analysis (CFA) was used to test the fit indices of BSMAS. The model fit of observed data (with Satorra-Bentler adjustment for non-normality data) was then assessed using many model fit indicators with respective cut-offs, including [33]: Relative Chi-square (χ^2^/df) ≤ 3.0; Root Mean Square Error of Approximation (RMSEA) ≤ 0.08; Standardized Root Mean Square Residual (SRMR) ≤ 0.08 for a good fit; and Comparative Fit Index (CFI) ≥ 0.9 for acceptable fit.

## 3. Results

### 3.1. Descriptive Characteristics

Table 1 shows the demographic characteristics of 173 participants. Most respondents were female (81.5%), about 85% of respondents were single, and 83.2% were living with family. Facebook, Zalo, and Youtube were the most popular social media platforms among Vietnamese youths (97.1%, 80.9%, and 75.7%, respectively). Talking to friends and updating news were the major purposes of using social media among participants (43.9% and 47.4%, respectively). The mean social media time per day was 5.9 h.

Figure 1 shows six characteristics of social media addiction among participants. Of the participants, 16.8% reported that they had often/very often tried to stop using social media but failed. About 16.2% of participants often/very often spent time thinking/planning to use social media platforms; 15% felt the need to use social media or used it to forget about personal problems; 11.5% of participants became anxious or agitated when they were banned. However, only 8.1% felt the negative impact of social media overuse.

### 3.2. Status of Social Media Addiction among Vietnamese Youths

Table 2 shows the average score of social media addiction by age group among Vietnamese youths. Overall, males had higher scores on social media addiction than females (15.3 and 14.5, respectively). Higher social media addiction scores were also prevalent in people living in rural or mountain areas or those who were single. Participants who used social media to play games also had higher addiction levels in both age groups.

### 3.3. Structural Validity of Bergen Social Media Addiction Scale

Figure 2 shows the scree plot and parallel analysis, which shows that the one-factor model fit with the current data. In the one-factor model, the value of factor loading was reported as 0.51 to 0.77 (Table 3). Furthermore, the mean score of each item of BSMAS ranged from 2.14 to 2.64 points. The Skewness and Kurtosis coefficients ranged from −0.03 to 0.42 and 2.15 to 3.35, respectively, showing an acceptable range and indicating the fairly symmetrical distribution. Moreover, all items of BSMAS showed high correlation coefficients with other items (r > 0.6). These results indicate that the items surveyed in the BSMAS effectively reflect the latent construct to assess social media addiction.

A confirmatory factor analysis was conducted on the six BFAS items and demonstrated acceptable goodness-of-fit (GOF) indexes for the one-factor model (Figure 3). In particular, the value of RMSEA was 0.044 (95%CI: 0.000; 0.102); CFI was 0.992; TLI was 0.987; and SRMR was 0.032.

### 3.4. Potential Predictors of Social Media Addiction

Table 4 presents a number of factors associated with the status of social media addiction among participants. It was suggested that social media addiction correlated with both scales. When the stress related to neglect by other users scale increased by 1 point, the score of social media addiction also increased by 0.035 points (Coef. = 0.35; 95%CI: 0.08; 0.62). The social media addiction score also increased by 0.08 when FOMO scores increased by 1 point.

## 4. Discussion

### 4.1. Principal Results

To our knowledge, this is one of the first studies to characterize the nature of social media usage and investigate the prevalence of social media usage among Vietnamese youths. While the majority of participants sampled were females, the results highlighted differences in the baseline demographics of the participants. In terms of the severity of the social media addiction, as assessed by the BSMAS, males were found to have greater severity of social media addiction as compared to females. Some factors, such as the locality of accommodation (rural versus mountainous areas) and relationship status also affected the overall scores. The results indicated Facebook, Zalo, and Youtube to be the most popular social media platforms among Vietnamese youths. Individuals who used social media for gaming also had higher BSMAS scores. This study also examined factors associated with BSMAS, such as perceived stress and fear of missing out.

### 4.2. Comparison with Prior Work

Our results suggested that social media addiction was more prevalent in males than females. Other studies, such as those by Alnjadat et al. (2019) [34] and Stănculescu et al. [35], also reported a high respondent rate among females and a higher percentage of males being addicted as compared to females. One potential explanation for this trend lies in the cultural dynamics of countries. In Middle Eastern countries such as Arab countries or in developing countries like Vietnam, females do not usually have an online social media profile, while males tend to use social networks to make friends. Indeed, in more developed countries, females tend to have more addictive use of social media involving elements of social interaction, whereas males tend to have a preference for more solitary online activities [10,36,37]. This study also demonstrated a higher prevalence of social media addiction amongst young individuals, which is consistent with previous research, such as that by Andressen et al. (2017) [38]. This has been attributed to the greater familiarity with internet use as more and more activities, both entertainment and education, are being switched to online platforms [39,40].

Another important finding of our study is the association between perceived stress and fear of missing out and the overall severity of the social media addiction scores. This trend is consistent with that of previous research works about perceived stress and problematic social networking use [41]. One of the identified factors associated with the relationship between perceived stress and social media usage was underlying depression and anxiety symptoms [41]. Individuals with greater levels of psychological resilience tend to have lower levels of psychological distress, depression, and anxiety, which correspond with lower social media addiction scores. Fear of missing out is defined as the pervasive apprehension that others might be having rewarding experiences when one is absent. Previous studies such as one by Fabris et al. (2020) [30] have reported an association between heightened stress associated with neglect by online peers and social media addiction. Interestingly, the impact of social media on FOMO is dual in nature. As technology provides a means of borderless and timeless communication, FOMO scores have indeed decreased for participants in other studies [42]. However, unlimited access to information also makes teenagers aware of many other existing sources of entertainment, education, and connections that they are unable to access. As a result, social media use has simultaneously exacerbated the severity of FOMO for teenagers. Therefore, more research is needed to find a solution to this vicious cycle and identify the line between effective use and overuse of social media.

### 4.3. Implications and Limitations

Several implications arise from this study. In terms of research, further studies are needed to examine the prevalence of social media addiction, and in particular, addiction to unique platforms. Longitudinal studies are also important in assessing the trends of social media addiction and the correlation between social media addiction and psychological well-being or comorbidities. As for clinical implications, diagnosis criteria or a standard clinical scale should be developed. Policymakers should also consider more intensive public education, potentially in schools, to educate children and adolescents regarding the risk of excessive internet and social media use and to provide education about how to manage excessive use behaviors. The increase in the popularity of social media means that overuse and addiction problems will soon surge in both quantity and severity. With the pending crisis of social media addiction, now is the time to address these issues in their infancy, rather than tackling the devastating consequences later.

The strength of this study lies in the diversity of the sample. We were able to recruit and investigate the prevalence of social media addiction amongst Vietnamese youth during the COVID-19 pandemic. The online survey model helped us reach participants across different regions and with different demographic characteristics, which increased the representativeness of the results.

However, several limitations remain. First, the sample size was limited, given that we recruited only youths who were under the age of 30 years old and who were affected by the COVID-19 pandemic. As the research was conducted online, and recruitment took place across 2 months, this limited the sample size and might have affected to some extent the representativeness of the overall sample as well as the results of this study. Only a small cross-section of youths was involved, which may impact this study’s degree of external validity. Therefore, a cohort study on social media addiction among a large group of youths could be conducted. However, efforts to mitigate this effect were undertaken by using a core group (comprising leaders from the youth unions in different public institutions, companies, and organizations) to reach out to their network of participants. Nonetheless, we do concur that one of the limitations was that we largely based our sampling on a convenience, online sample. Second, the results of this survey might be biased, as most questions asked about personal preferences and feelings. Lastly, while we investigated the association between perceived stress and fear of missing out, we were not able to take into account the existence of underlying psychological distress, which might influence the relationship between perceived stress and the severity of social media addiction. Nevertheless, this study was able to highlight alarming figures as well as suggested research, clinical, and policy implications.

## 5. Conclusions

This study lays a foundation for social media addiction research in Vietnam. Alongside technological breakthroughs, the prevalence of social media addiction and its consequences are also becoming more and more evident and undeniable. Our results show the need for public education in addressing excessive use as well as further research on trends in social media addiction and its association with other psychiatric disorders.

## Figures and Tables

**Figure 1 ijerph-19-14416-f001:**
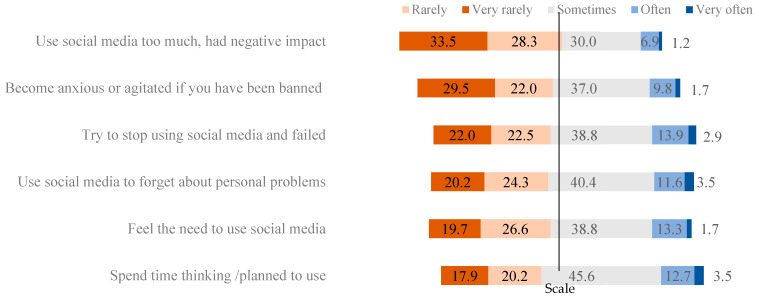
Social media addiction characteristics among Vietnamese youths.

**Figure 2 ijerph-19-14416-f002:**
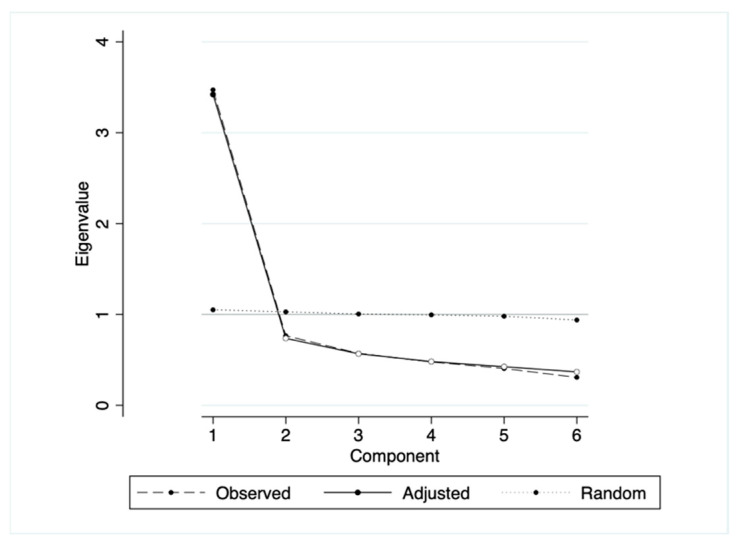
Scree parallel plot.

**Figure 3 ijerph-19-14416-f003:**
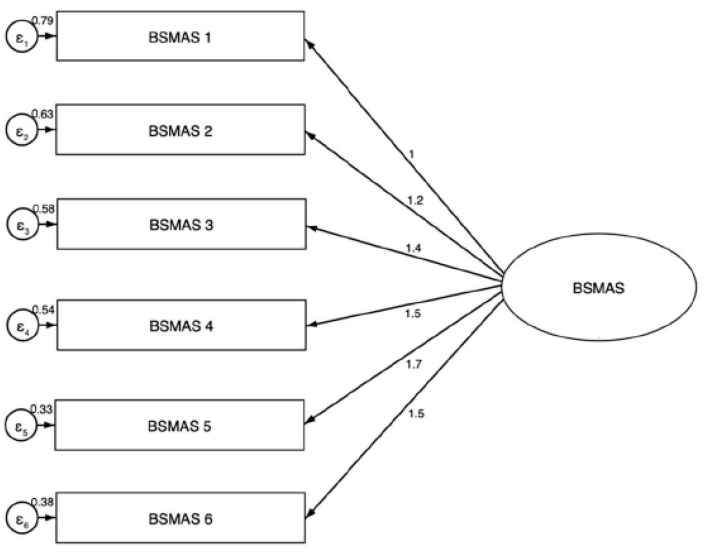
Factor structure and standardized loadings (CFA) of items of Bergen Social Media Addiction Scale. (RMSEA: 0.044 (95%CI: 0.000; 0.102); CFI = 0.992; TLI = 0.987; SRMR = 0.032).

**Table 1 ijerph-19-14416-t001:** Demographic characteristics of participants (*n* = 173).

Characteristics	Below 18 Years Old		18–24 Years Old		Above 24 Years Old		Total		*p*-Value
	n	%	n	%	n	%	n	%	
**Total**	36	20.8	109	63.0	28	16.2	173	100.0	
**Gender**									
Male	10	27.8	15	13.8	7	25.0	32	18.5	0.11
Female	26	72.2	94	86.2	21	75.0	141	81.5	
**Education**									
Below high school and high school	35	97.2	8	7.3	2	7.1	45	26.0	<0.001
College	1	2.8	25	22.9	4	14.3	30	17.3	
Tertiary and higher	0	0.0	76	69.7	22	78.6	98	56.7	
**Current location**									
Urban	23	63.9	51	46.8	5	17.9	79	45.7	<0.001
Suburban	2	5.6	16	14.7	9	32.1	27	15.6	
Rural/mountain area	11	30.6	42	38.5	14	50.0	67	38.7	
**Marital status**									
Single	32	88.9	107	98.2	8	28.6	147	85.0	<0.001
other (married/widowed/not want to share)	4	11.1	2	1.8	20	71.4	26	15.0	
**Currently living with**									
Family	35	97.2	83	76.2	26	92.9	144	83.2	<0.001
Others	1	2.8	26	23.9	2	7.1	29	16.8	
**The number of social networks used**									
1	6	16.7	6	5.5	2	7.1	14	8.1	0.12
≥2	30	83.3	103	94.5	26	92.9	159	91.9	
**Social networks used**									
Facebook	36	100.0	106	97.2	26	92.9	168	97.1	0.18
Zalo	20	55.6	94	86.2	26	92.9	140	80.9	<0.001
Youtube	29	80.6	87	79.8	15	53.6	131	75.7	0.01
Instagram	18	50.0	75	68.8	8	28.6	101	58.4	<0.001
Pinterest	14	38.9	17	15.6	2	7.1	33	19.1	<0.001
Snapchat	10	27.8	18	16.5	1	3.6	29	16.8	0.03
Twitter	9	25.0	19	17.4	0	0.0	28	16.2	0.01
**The main purpose of using social networks**									
Talk with friends	19	52.8	49	45.0	8	28.6	76	43.9	0.04
Update news	14	38.9	49	45.0	19	67.9	82	47.4	
Play games	3	8.3	2	1.8	0	0.0	5	2.9	
Other	0	0.0	9	8.3	1	3.6	10	5.8	
	**Mean**	**SD**	**Mean**	**SD**	**Mean**	**SD**	**Mean**	**SD**	***p*-Value**
**Age**	16.4	0.5	19.9	1.2	28.8	2.1	20.6	4.1	<0.001
**Years of use of social networks**	4.3	3.1	5.7	2.3	8.9	3.4	5.9	3.0	<0.001
**Time using social platforms/day** **(hour** **s** **)**	3.9	3.0	4.7	2.6	2.9	2.1	4.3	2.7	<0.001
**Number of social platforms** **used**	1.8	0.4	1.9	0.2	1.9	0.3	1.9	0.3	0.10
**Fear of missing out (FOMO Scale)** **(range: 10–50)**	28.4	10.7	26.0	8.3	22.0	9.6	25.8	9.2	0.02
**Stress associated with neglect and negative reactions by online Peers** **(range: 8–40)**	18.5	7.5	17.6	8.1	12.3	6.7	16.9	8.0	<0.001
Stress associated with neglect by other users (SSN) (range: 4–20)	8.9	4.1	8.2	4.2	6.0	3.5	8.0	4.1	0.02
Stress associated with negative reactions by other users (range: 4–20)	9.7	3.9	9.3	4.4	6.3	3.5	8.9	4.3	<0.001

**Table 2 ijerph-19-14416-t002:** Status of social media addiction among Vietnamese youths (*n* = 173).

	Social Media Addiction (Range: 6–30)											
	Below 18			18–24			Above 24			Total		
	Mean	SD	*p*-Value	Mean	SD	*p*-Value	Mean	SD	*p*-Value	Mean	SD	*p*-Value
**Total**	15.1	4.5		15.1	4.7		12.6	4.3		14.7	4.7	
**Gender**												
Male	16.3	4.5	0.42	16.0	4.7	0.25	12.4	5.1	0.63	15.3	4.8	0.33
Female	14.7	4.5		14.9	4.7		12.7	4.2		14.5	4.6	
**Education**												
Below high school and high school	15.3	4.5	0.23	13.8	4.9	0.01	14.5	9.2	0.14	15.0	4.7	0.20
College	10.0	0.0		12.8	4.5		17.3	5.1		13.3	4.8	
Tertiary and higher	0.0	0.0		15.9	4.5		11.6	3.2		15.0	4.6	
**Location**												
Urban	14.8	4.7	0.92	15.0	4.9	0.23	12.0	4.2	0.94	14.7	4.8	0.18
Suburban	16.0	2.8		13.4	5.6		12.6	2.2		13.3	4.6	
Rural/mountain area	15.7	4.6		15.8	4.0		12.9	5.4		15.1	4.5	
**Current living with**												
Family	15.0	4.6	0.39	14.6	4.4	0.12	12.8	4.4	0.19	14.4	4.5	0.12
Others	19.0	0.0		16.7	5.4		9.5	2.1		16.2	5.4	
**Marital status**												
Single	15.2	4.6	0.95	15.1	4.6	0.92	10.6	2.6	0.16	14.9	4.6	0.28
Other (married/widowed/not want to share)	14.8	4.3		13.5	10.6		13.4	4.6		13.6	4.8	
**The purpose of using social networks**												
Talk with friends	14.2	4.8	0.15	15.5	4.9	0.90	14.0	4.5	0.07	15.0	4.8	0.39
The news feed	15.3	3.6		14.7	4.3		11.5	3.6		14.1	4.2	
Play games	20.3	4.7		12.0	8.5		0.0	0.0		17.0	7.1	
Other	0.0	0.0		15.2	5.9		22.0	0.0		15.9	5.9	
**The number of social networks used**												
1	14.2	3.7	0.58	16.3	4.3	0.46	8.5	0.7	0.10	14.3	4.4	0.78
≥2	15.3	4.7		15.0	4.7		12.9	4.3		14.7	4.7	

**Table 3 ijerph-19-14416-t003:** Basic descriptions and reliability of Bergen Social Media Addiction Scale (BSMAS).

Items	Mean	SD	Factor Loading	Krewness	Kurtosis	Item–Total Correlation	Cronbach’s Alpha If Item Deleted
BSMAS 1	2.64	1.03	0.51	−0.03	2.57	0.63	0.86
BSMAS 2	2.51	1.01	0.63	0.04	2.32	0.72	0.83
BSMAS 3	2.54	1.05	0.68	0.13	2.49	0.75	0.83
BSMAS 4	2.53	1.07	0.73	0.07	2.28	0.78	0.82
BSMAS 5	2.32	1.06	0.81	0.19	2.15	0.84	0.81
BSMAS 6	2.14	1.00	0.77	0.42	3.35	0.80	0.82
BSMAS score (range: 6–30)	14.68	4.68		0.12	2.74		

(BSMAS 1: spend a lot of time thinking about social media or planning how to use it; BSMAS 2: spend a lot of time thinking about social media or planning how to use it; BSMAS 3: feel an urge to use social media more and more; BSMAS 4: use social media in order to forget about personal problems; BSMAS 5: have tried to cut down on the use of social media without success; BSMAS 6: become restless or troubled if prohibited from using social media).

**Table 4 ijerph-19-14416-t004:** Multivariate logistic regression for factors associated with social media addiction.

Characteristics	Social Media Addiction	
	Coeff.	95% CI
**Individual Characteristics**		
**Age group**		
Below 18	Ref.	
18–24	−1.48	−4.38; 1.43
Above 24	−2.96	−6.85; 0.93
**Education**		
Below high school and high school	Ref.	
College	1.06	−1.97; 4.08
Tertiary and higher	2.08	−0.74; 4.91
**Gender**		
Male	Ref.	
Female	−1.02	−2.80; 0.75
**Current living with**		
Family	Ref.	
Others	0.85	−1.01; 2.70
**Marital status**		
Single	Ref.	
Other (married/widowed/not want to share)	0.72	−1.90; 3.33
**Social Media Used**		
**The purpose of using social networks**		
Talk with friends	Ref.	
The news feed	−0.89	−2.31; 0.53
Play games	3.30	−0.96; 7.55
Other	0.11	−2.77; 2.99
**Number of social networks used**	0.46	−1.96; 2.88
**Years of use of social networks (years)**	0.03	−0.22; 0.28
**Time using social networks/day (hours)**	0.08	−0.18; 0.34
**Stress and Fear of Missing out Are Associated with Social Networks**		
**Stress associated with Neglect and Negative Reactions by Online Peers**		
Stress associated with neglect by other users (SSN) (Unit: one score)	0.35 ^b^	0.08; 0.62
Stress associated with negative reactions by other users (Unit: one score)	0.07	−0.19; 0.33
**Fear of missing out (FOMO Scale)** **(Unit: one score)**	0.08 ^a^	−0.01; 0.16

Note: (^a^: *p* < 0.1); (^b^: *p* < 0.05).

## Data Availability

Data are available upon request due to privacy restrictions.

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
