# Peer review of "Social Media Addiction among Vietnam Youths: Patterns and Correlated Factors"

_ijerph, 2022, doi:10.3390/ijerph192114416_

Round 1

Reviewer 1 Report

The paper presents a useful piece of research that was well conducted and clearly written. 

I have the following comments that I hope the authors will be able to address.

1) In section 2.2, I suggest adding a third level heading to improve the presentation of information.

2) Another limitation of the study is the sampling, please discuss it.

3) Reconsider the use of marital status in the analysis. What does 'others' mean? This is problematic, especially for participants under 18 years old.

4) The decision to categorise the sample into under 18, 18-24 and above 24 needs justification. Based on the presented results, I'm not entirely convinced that there is a need to have this three-category differentiation.

Author Response

The paper presents a useful piece of research that was well conducted and clearly written. 

I have the following comments that I hope the authors will be able to address.

  • In section 2.2, I suggest adding a third level heading to improve the presentation of information.

Author’s response: Thank you very much for your comments. I had added sub-heading of section 2.2.

  • Another limitation of the study is the sampling, please discuss it.

Author’s response: Thank you, Reviewer 1, for highlighting this point. We have discussed about the limitation of the study in the discussion. The amends are as follows, “The sample size was limited, given that we have recruited only youths who were under the age of 30 years old and who were affected by the COVID-19 pandemic. As the research was conducted online, and recruitment took place across 2 months, this limited the sample size, and might have affected to some extent the representativeness of the overall sample. Efforts to mitigate against this was undertaken, by using a core group (which comprised of leaders from the youth unions in different public institutions, companies and organizations) to reach out to their network of participants. Nonetheless, we do concur that one of the limitations was that we have largely based our sampling on a convenience, online sample.” (line 346-354)

  • Reconsider the use of marital status in the analysis. What does 'others' mean? This is problematic, especially for participants under 18 years old.

Author’s response: Thanks for your comments. In this study, our research subjects focused on adolescents and youths. Because the ranged age of participants was mostly focused on 16 to 24 and most of them reported a single status (85%). Therefore, only about 15% of the remaining participants reported other marital statuses such as married /widowed, and they were presented as the "other" group. I also added clearer information about "other" marital status in Table 1, Table 2, and Table 3 in this manuscript.

  • The decision to categorise the sample into under 18, 18-24 and above 24 needs justification. Based on the presented results, I'm not entirely convinced that there is a need to have this three-category differentiation.

Author’s response: Based on Law No. 53/2005/QH11 [1] of the Vietnam National Assembly, the youths included people from 16 to 30 years old. Specifically, people below 18 years old are grouped into "adolescents", 18-24 years old are grouped into "young adults", and the remaining group was from 25-30 years old. In this study, we followed the classification system based on the Law of the Vietnam National Assembly, and we also wish to examine for any difference among the three groups of participants.

Reviewer 2 Report

The topic is highly interesting as a social phenomenon and as a social problem.
The introduction structures main concepts properly, showing up a enough justification regarding how important this research is.
The text is written fluently and is easy to understand.
The method epigraph is correct and analysis techniques are appropriate.
Results are exposed solvent, and discussion and conclusion epigraphs too.
Nevertheless, not include results in the discussion (e.g. "48.6% as compared to 32%").
Review some typing mistakes and the format of references.

Author Response

The topic is highly interesting as a social phenomenon and as a social problem.
The introduction structures the main concepts properly, showing up enough justification regarding how important this research is.

1) The text is written fluently and is easy to understand.

Author’s response: Thank you very much for your comments.

2)The method epigraph is correct and analysis techniques are appropriate.

Author’s response: Thank you very much for your comments.

3)Results are exposed solvent, and discussion and conclusion epigraphs too.

Author’s response: Thank you very much for your comments.

4)Nevertheless, not include results in the discussion (e.g. "48.6% as compared to 32%").

Author’s response: Thank you for highlighting this. We have removed the presentation of any results in the discussion. The amends are “In terms of the severity of the social media addiction, as assessed by the BSMS, males were found to have greater severity of social media addiction as compared to females . Some factors, such as the locality of accommodation (rural versus mountainous areas), and relationship status also affected the overall scores. The results indicated Facebook, Zalo, and Youtube to be the most popular social media platforms among Vietnamese youths. Individuals who used social media for gaming also had higher BSMAS scores. This study also examined factors associated with BSMAS, such as perceived stress and fear of missing out.” (line 286-293)

The following paragraph has also been amended. “Our results suggested that social media addiction was more prevalent in males than females. Other studies, such as that by Alnjadat R et al. (2019) [2], and Stănculescu et al. [3], also reported a high respondent rate among females; and a higher percentage of males being addicted as compared to females.” (line 296-299).

5) Review some typing mistakes and the format of references.

Author’s response: Thank you very much for your comments. I edited references with mistakes.

Reviewer 3 Report

As the authors sad, the current study has the merit of investigating social media use in a culture where there are no previous studies on this topic. However, there are certain methodological problems that must be addressed. Please find details below:

[1] The bibliography can be updated. To include the most recent BSMAS validation studies based on advanced psychometric techniques (ie LPA, IRT) and which demonstrated its robustness.

Chen, I. H., Strong, C., Lin, Y. C., Tsai, M. C., Leung, H., Lin, C. Y., Pakpour, A. H., & Grifiths, M. D. (2020). Time invariance of three ultra-brief internet-related instruments: Smartphone Application-Based Addiction Scale (SABAS), Bergen social media addiction scale (BSMAS), and the nine-item internet gaming disorder scale-short form (IGDS-SF9) (Study part B). Addictive Behaviors, 101.

Cheng, C.,  Ebrahimi, O.V., &  Luk, J.W. (2022). Heterogeneity of prevalence of social media addiction across multiple classification schemes: Latent profile analysis. Journal of Medical Internet Research, 24 (1) (2022), p. e27000

Stănculescu, E. (2022). The Bergen Social Media Addiction Scale validity in a Romanian sample using item response theory and network analysis. International Journal of Mental Health and Addiction

 [2] The theoretical framing of the research should also be mentioned.

[3] You did not mention anything about the validity of the local language versions of the scales used in this study

[4] You calculated gender differences in social media addiction without highlighting measurement invariance across gender

[5] In fact, you did not give any information about the psychometric qualities of the adapted version of the BSMAS (construct validity)

[8] line 227 you can cite a recent paper that emphasized that females have higher risk of social media addiction

Stănculescu, E. & Griffiths, M.D. (2022). Social media addiction profiles and their antecedents using latent profile analysis: The contribution of social anxiety, gender, and age. Telematics and Informatics, 74, Article 101879

Author Response

As the authors said, the current study has the merit of investigating social media use in a culture where there are no previous studies on this topic. However, there are certain methodological problems that must be addressed. Please find details below:

[1] The bibliography can be updated. To include the most recent BSMAS validation studies based on advanced psychometric techniques (ie LPA, IRT) and which demonstrated its robustness.

Chen, I. H., Strong, C., Lin, Y. C., Tsai, M. C., Leung, H., Lin, C. Y., Pakpour, A. H., & Grifiths, M. D. (2020). Time invariance of three ultra-brief internet-related instruments: Smartphone Application-Based Addiction Scale (SABAS), Bergen social media addiction scale (BSMAS), and the nine-item internet gaming disorder scale-short form (IGDS-SF9) (Study part B). Addictive Behaviors, 101.

Cheng, C.,  Ebrahimi, O.V., &  Luk, J.W. (2022). Heterogeneity of prevalence of social media addiction across multiple classification schemes: Latent profile analysis. Journal of Medical Internet Research, 24 (1) (2022), p. e27000

Stănculescu, E. (2022). The Bergen Social Media Addiction Scale validity in a Romanian sample using item response theory and network analysis. International Journal of Mental Health and Addiction

Author’s response: We appreciate the references that the Reviewer has shared. We have carefully reviewed each of these suggestions and included them in the introductory and method sections paragraph, mainly to highlight the advances in the field. We had added “Since these initial studies, others such as Chen et al. (2020) [4] have examined the time invariance of three questionnaire instruments, that of the Smartphone Application-Based Addiction Scale (SABAS), Bergen Social Media Addiction Scale (BSMAS) and the nine-item Internet Gaming Disorder Scale-Short Form (IGDS-SF9) amongst Chinese students. The authors reported that the results obtained from these scales are time invariant across at least three months. In another recent study by Chang C et al. (2022) [5], the authors highlighted how the prevalence rates of social media addiction varies, and that those researchers who adopt a more polythetic approach tend to find there being more consistency in the prevalence rates they obtained from different samples. In addition, the adoption of the polythetic schematic approach allows for the identification of more members of the high-risk group. In another validation study by Stanculescu et al. (2021) [6], the psychometric properties of the scale has been validated amongst a Romanian sample, and aspects of the diagnostic criteria that were most relevant for the diagnosis of social media addiction amongst Romanian individuals were salience, conflict, withdrawal, and mood modification.” (line 96-109)

 [2] The theoretical framing of the research should also be mentioned.

Author’s response: Thank you very much for your comments. Before conducting this study, we reviewed a lot of previous studies associated with our topic to develop the theoretical framing as well as our questionnaire. In which, stresses associated with peers’ endorsement or rejection and fear or missing out were factors highly associated with internet addiction among young individuals. And in this study, “Therefore, our work aims to determine the prevalence of social media addiction amongst Vietnamese individuals, the factors associated with this addiction such as peers’ endorsement or rejection, fear of missing out and propose interventions accordingly.” I hope that with this clarification, there is now better understanding of the reasons as to why we have decided to embark and conduct this study. (line 121-124)

[3] You did not mention anything about the validity of the local language versions of the scales used in this study.

Author’s response: Thank you very much for your comments. I had added some information about the validity of BSMAS as follows: “Up until now, the psychometric properties of the BSMAS have been validated in Italian [7], English [8], Hong Kong [4, 9], Iran [10]” (Line 150-152).

To the best of our knowledge, there are no valididy of the Vietnamese version of BSMAS. In this study, based on the English version of this scale, we translated the instruments into Vietnamese. For the validation of the translated scale, we have added on more information in the methods sections, as follows,

“The survey was first piloted on several youths to ensure the cross-cultural validity of translated instruments in Vietnamese as well as to test the logical order and text of each question. After that, the revised questionnaire was uploaded to the online survey portal.” (line 137-141)

[4] You calculated gender differences in social media addiction without highlighting measurement invariance across gender.

Author’s response: Thank you very much for your comments. In this study, we aimed to determine the prevalence of social media addiction amongst a sample of Vietnamese youths as well as identify potential factors that might have associated with social media addiction. Furthermore, we did not design this study as a validation study of the BSMAS. Therefore, in this study, we only described and compared the differences in social media addiction regarding several potential characteristics such as gender, education, location, etc.

[5] In fact, you did not give any information about the psychometric qualities of the adapted version of the BSMAS (construct validity)

Thank you very much for your comments. Based on your suggestions, we calculated some indices to assess the construct validity of BSMAS. Based on the observed data, exploratory factor analysis (EFA) with principal component analysis (PCA) were utilized to identify the instrument's ideal structural model. The number of components was calculated using the scree plot and parallel analysis, as well as eigenvalues and the amount of variance explained. Items with a loading value ≥ 0.4 were considered to be included in the relevant component [11].

Then, Confirmation Factor Analysis (CFA) was used to test the fit indices of BSMAS. The model fit of observed data (with Satorra-Bentler adjustment for non-normality data) was then assessed using many models fit indicators with respective cut-offs, including [12]: Root Mean Square Error of Approximation (RMSEA) ≤ 0.08; Standardized Root Mean Square Residual (SRMR) ≤ 0.08 for a good fit and Comparative Fit Index (CFI)  ≥ 0.9 for acceptable fit. (line 197-208)

The results as follows: (line 239-261)

Figure 1: Scree Parallel Plot

Figure 1 showed the scree plot and parallel analysis, which showed that the 1-factor model fitted with the current data. In 1-factor model, the value of factor loading was reported at from 0.51 to 0.77 (Table 1). Furthermore, the mean score of each items of BSMAS ranged from 2.14 to 2.64 points. The Skewness and Kurtosis coffeeicients ranged from -0.03 – 0.42, and 2.15 – 3.35, respectively, showed a acceptable ranged to indicate the fairly symmetrical distribution. Moreover, all items of BSMAS showed a high correlation coefficiencets with other items (r>0.6). These results indicate that the items surveyed in the BSMAS effectively reflect the latent construct to assess the social media addiction.  

Table 1:Basic descriptions and reliability of Bergen Social Media Addiction Scale (BSMAS)

Items

Mean

SD

Factor loading

Krewness

Kurtosis

Item-total correlation

Cronbach's alpha if item deleted

BSMAS 1

2.64

1.03

0.51

-0.03

2.57

0.63

0.86

BSMAS 2

2.51

1.01

0.63

0.04

2.32

0.72

0.83

BSMAS 3

2.54

1.05

0.68

0.13

2.49

0.75

0.83

BSMAS 4

2.53

1.07

0.73

0.07

2.28

0.78

0.82

BSMAS 5

2.32

1.06

0.81

0.19

2.15

0.84

0.81

BSMAS 6

2.14

1.00

0.77

0.42

3.35

0.80

0.82

BSMAS score (range: 6-30)

14.68

4.68

0.12

2.74

(BSMAS 1: spend a lot of time thinking about social media or planning how to use it; BSMAS 2: spend a lot of time thinking about social media or planning how to use it; BSMAS 3: feel an urge to use social media more and more; BSMAS 4: use social media in order to forget about personal problems; BSMAS 5: have tried to cut down on the use of social media without success; BSMAS 6: become restless or troubled if you are prohibited from using social media.

Figure 2: Factor structure and standardized loadings (CFA) of items of Bergen Social Media Addiction Scale

(RMSEA: 0.044 (95%CI: 0.000; 0.102); CFI  = 0.992; TLI  = 0.987; SRMR  = 0.032)

A confirmatory factor analysis was conducted on the six BFAS items and demonstrated acceptable goodness-of-fit (GOF) indexes for the one-factor model. In particular, the value of RMSEA was 0.044 (95%CI: 0.000; 0.102); CFI  was 0.992; TLI  was 0.987; and SRMR  = 0.032.

[8] line 227 you can cite a recent paper that emphasized that females have higher risk of social media addiction

Stănculescu, E. & Griffiths, M.D. (2022). Social media addiction profiles and their antecedents using latent profile analysis: The contribution of social anxiety, gender, and age. Telematics and Informatics, 74, Article 101879

Author’s response: Thank you very much for your suggestion. I cited this paper in our discussion section (line 284).

Reviewer 4 Report

The study surveyed 173 young adults in Vietnam to determine the prevalence of social media addiction amongst Vietnamese individuals. This is a most needed study and with a well-developed introduction, however, when it comes to the method and results, it raised a few concerns. Here I tried to add a few points, but I recommend authors to use more appropriate tables and figures to report the findings. For example, this study's most important part is learning about social media addiction.

In the method section, one may be interested to see the cut point for BSMAS. If the score moves between 6-30, what is the cut point for showing the addiction, the authors said "higher scores indicated more severe social media addiction" higher than what? The same goes for stress scales. What is the cut point? Then if there is a cut point, I would compare the results between 'addicted' and 'non-addicted' by controlling demographic variables. I would also use box whisker plots to compare the distribution.

Here are a few more comments:

Abstract:

1. The abstract's method section is so highlighted that I encourage authors to rewrite the abstract by reporting four sections background (it is fine now), method, data, results, and conclusion; even if the journal requires a non-structural, these four sections should be covered.

2. Method, did you compute the sample size? How did you get to 173?

3. Time using social platforms/ day is measured by hours? Please add.

4. Table 2 is confusing; the title says the status of social media addiction. What are the numbers reported, for example, male 16.3 (4.5). What does it stand for?

5. Table 3, what is the dependent variable for this paper? Please clarify more in the method section by adding outcome variables.

6. In table 3, I consider the model's stress and fear variables as outcome variables to be tested.

  1. Discussion: the study compared media addiction among age-cat; however, this has not been highlighted in the discussion. I encourage authors to address my major comments (above) and then add more discussion (depends on the results).
  2. Some parts of the limitation sections belong to the discussion for example, lines 257-263.

Author Response

The study surveyed 173 young adults in Vietnam to determine the prevalence of social media addiction amongst Vietnamese individuals. This is a most needed study and with a well-developed introduction, however, when it comes to the method and results, it raised a few concerns. Here I tried to add a few points, but I recommend authors to use more appropriate tables and figures to report the findings. For example, this study's most important part is learning about social media addiction.

In the method section, one may be interested to see the cut point for BSMAS. If the score moves between 6-30, what is the cut point for showing the addiction, the authors said "higher scores indicated more severe social media addiction" higher than what? The same goes for stress scales. What is the cut point? Then if there is a cut point, I would compare the results between 'addicted' and 'non-addicted' by controlling demographic variables. I would also use box whisker plots to compare the distribution.

Author’s response: Thank you very much for your comments. I revised this sentence as follows “The total score ranged between 6 and 30, higher total scores indicating more severe social media addiction” (Line 48-151).

Furthermore, to the best of our knowledge, the cut-off point value of BSMAS was only identified in some versions of BSMAS such as the Romanian version [13], or the Chinese version [14], and not including the Vietnamese version. The original version of BSMAS used the total score to assess the level of severe social media addiction. Therefore, in this study, we applied the total score instead of the cut-off point of this scale.

Here are a few more comments:

Abstract:

  • The abstract's method section is so highlighted that I encourage authors to rewrite the abstract by reporting four sections background (it is fine now), method, data, results, and conclusion; even if the journal requires a non-structural, these four sections should be covered.

Author’s response: Thank you very much for your comments. I had rewritten the abstract of this manuscript to report four sections: background, method, results, and conclusion. (line 20-line 38)

  • Method, did you compute the sample size? How did you get to 173?

Author’s response: Thank you very much for your comments. I had added information about sample size as follows: “To calculate the sample size of this study, we used formula for estimating a population mean. With the expected mean score of BSMAS among adolescents = 15.24; standard deviation = 4.83 (according to a previous study in Iran [10]); confidence level = 95%, and relative precision = 0.05. After calculating, the necessary sample size was 155 participants. To prevent incomplete responses or dropout, 15% of the sample size was added, thus resulting in a total of 179 participants who were invited to participate in the study. At the end of the data collection period, the total of participants who agreed to involve in this study and completed the questionnaire was 173, with a response rate was 96.6%.” (line 123-130).

  • Time using social platforms/ day is measured by hours? Please add.

Author’s response: Thank you very much for your comments. I had added the unit of this variable in line 167 and Table 1.

  • Table 2 is confusing; the title says the status of social media addiction. What are the numbers reported, for example, male 16.3 (4.5). What does it stand for?

Author’s response: Thank you very much for your comments. In Table 2, based on The Bergen Social Media Addiction Scale (range: 6-30 points), we want to describe in detail about the level of severe social media addiction among participants regarding the age group and some other variables such as gender, education, location, etc. For example, the mean score of BSMAS among males who were below 18 years old was 16.3 (SD = 4.5).

  • Table 3, what is the dependent variable for this paper? Please clarify more in the method section by adding outcome variables.

Author’s response: Thank you very much for your comments. In this study, social media addiction based on The Bergen Social Media Addiction Scale (range: 6-30 points) was identified as the dependent variable. Furthermore, I added the heading “2.3.1. Outcome variable”  in the method section to clarify about the research’s dependent variable (line 127).

  • In table 3, I consider the model's stress and fear variables as outcome variables to be tested.

Author’s response: Thank for your comments. In this study, we aimed to determine the prevalence of social media addiction amongst a sample of Vietnamese youths as well as identify potential factors that might have associated with social media addiction. And by reviewing a lot of previous study, we reliazed that stress and fear of missing out were potential factors associated with this topic. Therefore, to address objective of this study, total score of BSMAS was applied as a outcome variables.

Discussion:

  1. the study compared media addiction among age-cut; however, this has not been highlighted in the discussion. I encourage authors to address my major comments (above) and then add more discussion (depends on the results).

Authors’ response: Thank you very much for your comments.

  1. Some parts of the limitation sections belong to the discussion for example, lines 257-263.

Thank you for the suggestions. We have stated the title as Implications and limitations. The first part of the paragraph covers the implications of our work.

Round 2

Reviewer 1 Report

Thank you for addressing the comments shared. My third and fourth comments remain inadequately addressed.

As far as I'm able to gather, the legal marriage age is 18 for females and 20 for males in Vietnam. So having participants who are either married or widowed at 16 or 17 years old is problematic and raises ethical concerns. I strongly suggest excluding participants under 18 in this case. This is also related to how you categorise the age groups.

Author Response

Dear Reviewer 1, 

Thank you very much for your comments and suggestion. I would like to answer some of your comments as follows: 

1) As far as I'm able to gather, the legal marriage age is 18 for females and 20 for males in Vietnam. So having participants who are either married or widowed at 16 or 17 years old is problematic and raises ethical concerns. I strongly suggest excluding participants under 18 in this case. This is also related to how you categorise the age groups.

Author’s response: Thank you very much for your comments and suggestion. As you said, we confirmed that the legal marriage age in Vietnam for females is 18 years old or above, and for males is 20 years old or above, and having participants who are either married or widowed at 16 or 17 years old is problematic. But in this study, there were no people below 18 years old having married or widowed status. Actually, in this study, the "other" group in marital status not only included married or widowed people but also those who did not want to share their marital status (choose the "other, not want to share" option in this question). I would like to detail describe our question and all answer options that were used to explore the marital status of participants in this study as follows:  

Furthermore, I revised information about the "other (married /widowed/ not want to share)" option of the marital status variable to more clearly for this variable (Table 1, Table 2, and Table 4).

Moreover, the main objective of this study focused on social media addiction among youths. Also based on the Youth Law No. 53/2005/QH11 of the Vietnam National Assembly, youths included all people from 16 to 30 years old. Therefore, in this study, we want to include all participants from 16 or above to analyze.

2) The decision to categorise the sample into under 18, 18-24 and above 24 needs justification. Based on the presented results, I'm not entirely convinced that there is a need to have this three-category differentiation.

Author’s response: Based on Law No. 53/2005/QH11[1]  of the Vietnam National Assembly, the youths included people from 16 to 30 years old. Specifically, people below 18 years old are grouped into "adolescents", 18-24 years old are grouped into "young adults", and the remaining group was from 25-30 years old.

Furthermore, some previous studies also indicated that among youth generation, people in 16-18 years old (High school students) had the highest level of social media addiction, followed by 18-24 years old (university students), and above 24 years old [2-4]. To the best of our knowledge, up until now, in Vietnam, applied structured scale (eg., BSMAS) to investigate the situation of social media addiction among youths is limited. Therefore, in this study, we want to conduct a comprehensive study to explore the social media addiction situation among youths of a large range of ages according to the Youth Law of the Vietnam National Assembly (16-30 years old).

Reviewer 4 Report

Thank you for addressing my comments, no further comment.

Author Response

Dear Reviewer, 

Thank you for your comments and suggestions that allowed us to improve the quality of the manuscript greatly.

Round 3

Reviewer 1 Report

Thank you for the detailed explanation. Please include your explanations in the manuscript, in the method section. It should be clearly explained

1) why 16-30 year olds were recruited in section 2.1, and 

2) the rationale of the age bands and 'others' of the marital status in section 2.3.2.1

Author Response

Thank you for the detailed explanation. Please include your explanations in the manuscript, in the method section. It should be clearly explained

Author's response: Thank you for your comments and suggestions that allowed us to improve the quality of the manuscript greatly. 

1) Why 16-30 year olds were recruited in section 2.1.

Author’s response: Thank you very much for your suggestion. I had added the reason why people from 16-30 years old were recruited in section 2.1 as follows: (Line 129)

Individuals were eligible to participate in the study if they were (1) between 16 to 30 years old (based on Youth Law No. 53/2005/QH11 [25] of the Vietnam National Assembly); (2) currently residing in Vietnam and (3) capable of providing informed consent. 

2) The rationale of the age bands and 'others' of the marital status in section 2.3.2.1

Author’s response: Thank you very much for your comments. I added the rationale of age group and marital status in section 2.3.2.1 as follows:

2.3.2.1. Socioeconomic characteristic

Participants were asked about gender, age, household economy (rich/medium/poor), marital status (single/living together as spouse/married /widowed/not want to share), education (below high school and high school/college/tertiary and upper); current living location (urban/town/rural or mountain area); current living partner (family/others).

Furthermore, based on age, the participants were grouped into three subgroups. Specifically, people below 18 years old are grouped into "adolescents", 18-24 years old are grouped into "young adults", and the remaining group was from 25-30 years old [25, 30, 31]. (line 171-178).